# Risk Assessment and Response Strategy for Pig Epidemics in China

**DOI:** 10.3390/vetsci10080485

**Published:** 2023-07-26

**Authors:** Zizhong Shi, Junru Li, Xiangdong Hu

**Affiliations:** 1Institute of Agricultural Economics and Development, Chinese Academy of Agricultural Sciences, Beijing 100081, China; shizizhong@caas.cn; 2College of Economics and Management, China Agricultural University, Beijing 100083, China; lijunru901@163.com

**Keywords:** pig, animal disease, African swine fever, risk assessment, policy recommendations

## Abstract

**Simple Summary:**

The frequent occurrence of pig epidemics has restricted the sustainable and healthy development of the pig industry and its ability to enhance the supply of pork, negatively impacting China’s economic and social development. In recent years, China has faced many pig epidemic risks and challenges, with one high-risk area and two medium–high-risk areas. The epidemic risk was highest in Beijing, Hainan, Liaoning, Tibet and Zhejiang. This study will help to better prevent and mitigate epidemic risks, promote high-quality development of the pig industry, and meet the nutritional needs of residents.

**Abstract:**

Strengthening the analysis and risk assessment of the pig epidemic will help to better prevent and mitigate epidemic risks and promote the high-quality development of the pig industry. Based on a systematic understanding of live pig epidemics, a risk assessment index system was constructed, and the spatial and temporal variation characteristics of pig epidemics in China were explored by the entropy method. In recent years, the overall trend in pig epidemics over time first increased and then decreased; in space, the acceleration of the spread of epidemics across the country weakened. China still faces challenges, including many types and a wide range of diseases, large total livestock breeding and weak epidemic prevention and control capacity, and a large risk of introduced foreign animal epidemics. The spatial and temporal variations in the pig epidemic risk were obvious; one high-risk area, two medium–high-risk areas and 10 medium-risk areas have been found in recent years, during which time, the epidemic risk was highest in Beijing, Hainan, Liaoning, Tibet and Zhejiang. However, there were significant differences in the regional distribution of the risk level of pig epidemics in different years. To further build a secure “defense system” for the high-quality development of the pig industry, it is recommended to improve the monitoring and early warning system of pig epidemic risk, perfect the pig epidemic prevention and control system, and strengthen the regional collaboration mechanism of epidemic prevention and control.

## 1. Introduction

Accelerating the high-quality development of the pig industry and ensuring the stable and orderly operation of the pork market is an important means of meeting the animal-based nutrition needs of residents and safeguarding national food security. This is a basic guarantee for enhancing the level of regional economic and social development, improving the ability of farmers to increase production and income and forging a realistic path for accelerating the construction of an agricultural powerhouse and realizing people’s aspirations for a better life. In recent years, the frequent occurrences of African swine fever and other major animal epidemics have restricted the sustainable and healthy development of the pig industry and its ability to enhance the supply of pork, negatively impacting national economic and social development and the lives of urban and rural residents. Although, currently, incidences of African swine fever and other major animal epidemics have been alleviated, sporadic occurrences, as well as epidemic prevention and control, cannot be ignored. Data from the Ministry of Agriculture and Rural Affairs of the People’s Republic of China showed that 15 new outbreaks of African swine fever occurred nationwide in 2021, and 4500 pigs were killed and disposed of. Since then, China’s government has attached great importance to the construction of national biosafety risk prevention, control and management systems; it has become inevitable and necessary to carry out risk assessment and systematic research on major animal disease prevention and control as an important part of the national biosafety risk prevention and control system.

The literature on animal disease risk is primarily focused on risk assessment, animal disease prevention and control, etc. [1]. In terms of assessment methods, many research methods, such as Monte Carlo simulation, the entropy method, fuzzy integrated evaluation, semiquantitative risk assessment techniques, entity-relationship models, and the construction of risk assessment index systems, have been explored and proposed [2,3,4,5,6,7,8]. Moreover, Vos et al. compared and analyzed several generic risk assessment tools for animal disease transmission [9]. Squarzoni-Diaw et al. suggested that qualitative risk assessment can be an important tool when data are scarce [10]. Stärk and Salman emphasized the important role of surveillance and monitoring systems in risk assessment [11]. The risk areas of African swine fever were concentrated in eastern and southwestern China [12]. The risk of pig epidemics was mainly distributed among different links of the industrial chain, such as breeding, distribution and slaughter [13,14].

The issues of domestic animal disease transmission risk and international trade disease transmission risk have also been explored in many studies [15,16,17]. Studies have found that the risk of the introduction of African swine fever and classical swine fever into the United States through legal imports of pigs and their products is low [18]. The current import quarantine policy implemented by the Korean government has been effective in preventing the introduction of African swine fever into the country through the legal imports of pigs and their products [19]. The risk level of African swine fever being introduced into Japan varies among experts, with foreign experts being more pessimistic than Japanese domestic experts; with the reduction in the number of air travelers and restaurant food waste in China, the risk of African swine fever entering Japan through illegal imports of pig products from China has gradually decreased [20,21]. In one particular year, African swine fever entered countries in southeastern Europe with a high risk of transmission; however, there was a low probability of transmission to EU member states outside this region [22]. Previous studies have also assessed the risk of coronavirus disease 2019 (COVID-19), which has important implications for the risk assessment of major animal diseases in terms of perspectives and methodologies, such as geographic information system (GIS)-based spatial analysis techniques, analytic hierarchy processes, integrated weighting methods, “movement–contact” spatial interaction networks, and susceptible–exposed–infectious–removed (SEIR) models [23,24,25,26].

Other studies have suggested that to achieve the stable and orderly development of the pig industry, the important role of epidemic prevention and control must be given high priority in building a long-term mechanism [27,28]. From the perspective of epidemics and their risk prevention and control measures, there have been many discussions and systematic reviews in the domestic and foreign literature, including those on quarantine, isolation, immunization, killing and disposal, monitoring, subsidies, biosafety disposal, etc. [29,30,31]. China has a variety of policy measures for epidemic prevention and control, but frequent international trade, a high risk of transmission, and inadequate traceability systems make prevention and control more difficult [32]. Different studies had different views on the effectiveness of the implementation of epidemic prevention and control measures. A comparison of multiple porcine epidemic diarrhea control strategies found that intensive biosecurity measures were the most effective in reducing overall losses, and vaccination was the least cost-effective method [33]. Additionally, the implementation of biosecurity measures was more effective than animal health programs [34]. Killing and disposal measures were also effective in reducing economic losses from the epidemic outbreak [35]. When comparing killing and disposal measures and emergency vaccination programs used to control classical swine fever in the EU, the direct costs of emergency vaccination were lower, but the indirect costs were higher, and political intervention was an important reason why emergency vaccination measures were not economically attractive [36].

In general, with the current focus on African swine fever, the research that specifically analyzes this situation and risk assessments of pig epidemics needs further systematic and in-depth review. In particular, after African swine fever outbreaks, it is of great practical importance to review the circumstances of the pig epidemic, systematically explore the spatial and temporal variation characteristics associated with the risk level of the pig epidemic, and summarize the lessons learned to better improve the epidemic prevention and control system, prevent and resolve the epidemic risk, and promote the high-quality development of the pig industry in the future. Considering the shortcomings of existing studies, this study attempted to construct a pig epidemic risk assessment index system based on an understanding of pig epidemic circumstances starting in 2017; use the entropy method to assess the epidemic risk and understand the spatiotemporal characteristics; and, finally, propose corresponding countermeasure suggestions for reference.

## 2. Analysis of the Pig Epidemic Situation

### 2.1. Temporal Characteristics of Pig Epidemics

Animal epidemics are an important factor that restricts the smooth and orderly development of the pig industry; epidemics with greater impacts mainly include porcine reproductive and respiratory syndrome and African swine fever. Regarding the trend of pig epidemics in recent years, after 2018, affected by African swine fever, the number of cases, deaths, and cullings increased sharply. Afterward, as the epidemic prevention and control situation continued to improve, the pig epidemic gradually slowed down (Table 1 and Figure 1). In terms of different epidemic types, African swine fever has been the most serious in recent years, with the highest number of cases and deaths in 2019, and the highest number of cullings in 2018. In addition, in recent years, the more serious pig diseases have included swine erysipelas, swine pasteurellosis, porcine reproductive and respiratory syndrome and classical swine fever; other pig diseases, such as foot and mouth disease and porcine cysticercosis, were relatively rare. The number of cases of swine erysipelas reached a high level in 2017, and the number of deaths and cullings reached high levels in 2018. Swine pasteurellosis was the most serious disease in 2021, porcine reproductive and respiratory syndrome was relatively more serious in 2019, and classical swine fever was relatively more serious in 2018 (Table 1). In general, the situation of pig epidemics in recent years has been more complex, with African swine fever being the most severe. However, after the African swine fever epidemic, pig epidemics generally slowed down. The main reason was that the prevention and control of African swine fever has changed from “immaturity” to “maturity”, and China’s government has explored many measures, including zoning prevention and control.

### 2.2. Spatial Characteristics of Pig Epidemics

Before the African swine fever outbreaks, pig epidemics were mainly distributed in Jiangxi, Hubei, Chongqing and Guangxi provinces, while the number of cullings was generally at a relatively low level. After the outbreak of African swine fever, the regional distribution of the epidemic changed significantly, with concentrated outbreaks in various regions. In 2018, the top three provinces in terms of cases were Jiangxi, Yunnan and Sichuan, the top three provinces in terms of deaths were Liaoning, Yunnan and Chongqing, and the top three provinces in terms of cullings were Liaoning, Hunan and Fujian. In 2019, the regional distribution of the epidemic changed further, with the top three provinces in terms of cases and deaths being Guangxi, Sichuan and Heilongjiang, and the top three provinces in terms of cullings being Heilongjiang, Jiangsu and Guangxi. Since 2020, with the African swine fever epidemic under better control, the national pig epidemic situation has continued to improve, and only Gansu Province’s pig epidemic has been relatively serious. Overall, the regional distribution of pig epidemics in recent years was mainly affected by African swine fever, and with the African swine fever epidemic under control, pig epidemics have rapidly slowed nationwide. Currently, pig epidemics occur only sporadically in some provinces (Figure 2).

### 2.3. Risks and Challenges of Pig Epidemics

At present, although African swine fever and other major animal diseases are better controlled, the associated risks and challenges are still prominent. First, pig epidemics are diverse and widespread. According to the “List A, B and C Diseases” issued by the Ministry of Agriculture and Rural Affairs of the People’s Republic of China in 2022, there are currently a wide variety of pig diseases in China, such as A diseases, including African swine fever, foot and mouth disease, and swine vesicular disease. At present, a large number of pig diseases occur sporadically, and complete elimination is still far off. In terms of the speed and scope of the spread, African swine fever, for example, was first diagnosed in Liaoning Province in August 2018, and then the epidemic spread rapidly across the country within a short period of time, tremendously and negatively impacting the pig industry and market as well as the economy and society. The pig slaughters in 2019 fell by 21.6% compared with the previous year, and pork production dropped by 21.3%.

Second, the total amount of livestock breeding was large, and the epidemic prevention and control capacity was not strong. China is the world’s largest producer and consumer of pigs, and since the outbreak of African swine fever in 2018, China’s government has implemented a series of policy initiatives to accelerate the recovery of pig production to normal levels. The inventory of pigs reached 452.56 million heads in 2022, of which the inventory of breeding sows reached 43.90 million heads. Although the level of scale, intensification and modernization of pig breeding has improved, there is still a large gap between the expected goal of high-quality development and the levels of developed countries in terms of epidemic prevention and control capabilities. There are still more small and medium-sized farms, retail households and other business entities in which the awareness and ability regarding epidemic prevention and control are not strong, and the level of epidemic prevention and control of pig households needs further improvement. In addition, the current grassroots epidemic prevention and control system is not sound, the epidemic prevention team is relatively weak, and loopholes in quarantine supervision and other issues remain prominent. In China, where people prefer hot fresh meat, the long-distance transport and slaughter of live pigs also create uncertainty about the spread of pig epidemics.

Moreover, the risk of introduction of foreign animal diseases is greater. At present, the global situation of pig epidemics is still not optimistic. Data from the World Organization for Animal Health (WOAH) showed that in 2022, there were 6026 new outbreaks of African swine fever worldwide, with 159.80 thousand newly confirmed cases, 199.90 thousand heads killed and disposed of, and 38.50 thousand deaths. Although China has been strictly controlling foreign animal epidemics for a long time, including with the development and implementation of the “Emergency Response Plan for Major Animal Epidemics Inside and Outside China” and “Interim Measures for Veterinary Health Management of Animal Isolation and Quarantine Sites at Ports” and other policy initiatives, it still failed to block the introduction of animal diseases such as African swine fever into China. It is necessary to pay great attention to the variety of global animal diseases, the complexity of the pathogens and the seriousness of these situations and other issues; to learn from the current round of African swine fever regarding domestic exposures; to accelerate sound domestic animal disease prevention and control systems; and to reduce the risk of foreign disease introduction to protect the high-quality development of the pig industry.

## 3. Methods and Materials

### 3.1. Methods

To systematically assess the risk of pig epidemics and explore their spatial and temporal risk characteristics, this study constructed an evaluation index system and combined it with the entropy method. In defining the concept of risk assessment, the United Nations Office for Disaster Risk Reduction considers that risk consists of two parts: hazard and vulnerability [37]. Combining the domestic pig industry and epidemic situation, based on the principle of basic data and indicator availability, this study selected three indicators to measure the hazard of pig epidemics, including the morbidity rate, mortality rate and culling rate, and three indicators to measure the vulnerability of pig epidemics, including breeding density, industrial structure and prevention and control foundation. The pig epidemic risk assessment index system and its calculation methods are shown in Table 2.

Specifically, the morbidity rate was measured by the ratio of the number of cases caused by the epidemic to the number of pigs, the mortality rate was measured by the ratio of the number of pig deaths caused by the epidemic to the number of cases, and the culling rate was measured by the ratio of the number of pigs culled during the epidemic to the number of pigs. The higher the three indicators were, the more serious and dangerous the pig epidemic was. In particular, unlike morbidity and mortality rates, the culling rate was more indicative of the severity of the epidemic, such as the occurrence of African swine fever in recent years, which caused a higher culling rate, and although the realistic mortality rate was not high, a large number of pigs were culled due to factors such as the high lethality and rapid spread of African swine fever.

Breeding density was measured by the ratio of pig inventory to grain cultivation area. Pigs are grain-fed livestock, and a higher breeding density indicates a higher local pig carrying capacity and a higher epidemic risk faced by pig breeding. Industrial structure was measured by the ratio of pig industry output value to total agricultural output value. The higher the proportion of pig industry output value is, the more likely the pig industry is the leading industry in the province, the higher the corresponding scale of standardization and modernization, and the lower the epidemic risk faced by pig breeding. The prevention and control foundation was measured by the ratio of pig inventory to the number of people on staff in township animal husbandry and veterinary stations. The higher the ratio of pig inventory to the number of staff in the township animal husbandry and veterinary station is, the more pigs a single animal husbandry and veterinary staff member has to manage and serve, the weaker the force of pig epidemic prevention and control, the lower the intensity of epidemic prevention and control, and the greater the risk of an epidemic affecting pig breeding.

Considering the differences in different indices in terms of their outlines and positive and negative directions, this study first standardized the indices, then calculated the index weights at all levels by the entropy method, and finally combined the standardized index values and weights to calculate the pig epidemic risk index. It should be noted that the morbidity rate, mortality rate, culling rate, breeding density and prevention and control foundation were all positive indicators, while industry structure was a negative indicator. The standardized treatment formula is specified as follows:(1)$$x_{ij}=\frac{a_{ij}-\min\{a_{ij}\}}{\max\{a_{ij}\}-\min\{a_{ij}\}}$$
(2)$$x_{ij}=\frac{\max\{a_{ij}\}-a_{ij}}{\max\{a_{ij}\}-\min\{a_{ij}\}}$$
where Equation (1) is the normalization formula for positive indicators and Equation (2) is the normalization formula for negative indicators. *x_ij_* is the value of hazard and vulnerability indicators after normalization, *a_ij_* is the original value of each hazard and vulnerability indicator, and max{*a_ij_*} and min{*a_ij_*} are the maximum and minimum values of each hazard and vulnerability indicator, respectively.

Before calculating the weights of each indicator through the entropy method, the weight of the corresponding indicator of the *i*th region under the *j*th indicator needs to be calculated as follows:(3)$$p_{ij}=\frac{x_{ij}}{\sum_{i=1}^{n}x_{ij}}$$
where *p_ij_* is the corresponding indicator weight. Then, the entropy value of the *j*th indicator is calculated:(4)$$e_{j}=-k\sum_{i=1}^{n}p_{ij}\ln p_{ij}$$
where *e_j_* is the entropy value of the corresponding indicator and satisfies *e_j_
*≥ 0, *k* = 1/ln*n*. On this basis, the weights of each indicator are calculated as follows:(5)$$w_{j}=\frac{d_{j}}{\sum_{j=1}^{m}d_{j}}$$
where *w_j_* is the corresponding indicator weight, *d_j_* is the information entropy redundancy of the *j*th indicator, and *d_j_
*= 1 − *e_j_*.

### 3.2. Materials

The data for this study were obtained from the Official Veterinary Bulletin of the Ministry of Agriculture and Rural Affairs of the People’s Republic of China, China Animal Husbandry and Veterinary Yearbook, and China Statistical Yearbook. Considering that the Official Veterinary Bulletin from the Ministry of Agriculture and Rural Affairs of the People’s Republic of China was available only for October 2021 and earlier, the pig epidemic data for 2021 were only for January–September. Therefore, the time interval examined in this study was set as 2017–2020, where 2017 was the normal year before the outbreak of African swine fever, 2018–2019 was the most serious period of African swine fever, and 2020 was the year when the epidemic was better controlled. Overall, the time interval selected in this study covered the current round of the pig epidemic, and the pig industry in each region and its impact could better reflect the corresponding epidemic risk prevention and control capacity and its problems.

## 4. Results and Discussion

### 4.1. Results

Combined with the basic data, the indices of pig morbidity, mortality and culling rates, breeding density, industrial structure, and prevention and control foundation from 2017 to 2020 were calculated, and the risk of a pig epidemic and its hazard and vulnerability indices were calculated after standardization. Table 3 shows the risk of the pig epidemic and regional distribution obtained based on the mean value of the basic data from 2017 to 2020, and Figure 3 shows the regional distribution of the risk of a pig epidemic from 2017 to 2020. It should be noted that since the risk assessment data were based on the current year’s values, the risk shown in Figure 3 is an annual relative concept that reflects only the current year’s risk distribution and has no size reference value for cross-year comparisons but has reference for the current year’s risk distribution and cross-year risk regional changes.

Specifically, the hazard and vulnerability weights calculated by the entropy method for 2017–2020 were 0.679 and 0.321, respectively; the hazard indicators were 0.243, 0.105 and 0.652 for morbidity, mortality and culling rates, respectively; and the vulnerability indicators were 0.393, 0.125 and 0.482 for breeding density, industrial structure and prevention and control foundation, respectively. The weights of each indicator reflected that the risk of a pig epidemic came more from hazards, the culling rate regarding hazard factors was more important, and the foundation of prevention and control of vulnerability factors was more important. Regarding the characteristics of the regional distribution of pig epidemic risk, the top five provinces were Beijing, Hainan, Liaoning, Tibet and Zhejiang, with risk indices of 0.682, 0.367, 0.337, 0.282 and 0.268, respectively. The bottom five provinces in terms of risk were Shandong, Hebei, Jilin, Ningxia and Inner Mongolia, with risk indices of 0.097, 0.104, 0.112, 0.118 and 0.125, respectively.

In terms of hazards, the top five provinces were Beijing, Liaoning, Tibet, Qinghai and Jiangxi, with hazard indices of 0.799, 0.361, 0.329, 0.281 and 0.266, respectively. The bottom five provinces were Shandong, Guangdong, Henan, Hebei and Guizhou, with risk indices of 0.006, 0.017, 0.052, 0.052 and 0.07, respectively. Among them, the top provinces in terms of morbidity rates were Jiangxi, Chongqing, Qinghai, Tibet and Zhejiang, the top provinces in terms of mortality rates were Jilin, Qinghai, Inner Mongolia, Liaoning and Heilongjiang, and the top provinces in terms of culling rates were Beijing, Liaoning, Tibet, Tianjin and Xinjiang.

In terms of vulnerability, the top five provinces were Hainan, Fujian, Guangdong, Guangxi and Zhejiang, with vulnerability indices of 0.962, 0.566, 0.460, 0.439 and 0.438, respectively. The bottom five provinces were Jilin, Inner Mongolia, Heilongjiang, Shanxi and Ningxia, with vulnerability indices of 0.118, 0.122, 0.147, 0.148 and 0.149, respectively. The top provinces in terms of breeding density were Hainan, Fujian, Beijing, Guangdong and Guangxi; the top provinces in terms of industrial structure were Hunan, Yunnan, Sichuan, Anhui and Chongqing; and the top provinces in terms of prevention and control foundations were Hainan, Henan, Zhejiang, Anhui and Fujian. In terms of the overall situation of the risk of pig epidemics in different areas in recent years, there was 1 high-risk area, 2 medium–high-risk areas, 10 medium-risk areas, 17 medium–low-risk areas and 1 low-risk area nationwide. The risk of pig epidemics and regional prevention and control should not be ignored.

Regarding the trend in temporal changes in the risk of pig epidemics, the regional changes in the risk of pig epidemics in different years were obvious. Before the outbreak of African swine fever in 2017, the provinces with the highest risk of pig epidemics were Qinghai, Jiangxi and Chongqing; the risk was also relatively high in northwest, southwest and southeast China; and the risk was low in northeast and north China. There were three high-risk areas, no medium–high-risk areas, and three medium-risk areas, and the others were medium–low- and low-risk areas. In 2018, with the outbreak of African swine fever, the regional distribution of pig epidemic risk changed significantly, mainly concentrated in Beijing and Liaoning, with Zhejiang, Tianjin and Jiangxi also at relatively high levels, and the overall risk of pig epidemics shifted to the east. There was one high-risk and medium–high-risk area and three medium-risk areas, and the remaining areas were low-risk and medium–low-risk areas.

The year 2019 was a more severe year for the occurrence of African swine fever, and the risk of pig epidemics spread faster across the whole country. The highest risk of epidemics shifted to Tibet and Jiangsu; the epidemic risks in Qinghai, Xinjiang and Heilongjiang were also relatively high. There were two high-risk areas, three medium–high-risk areas, and two medium-risk areas, and the remaining areas were low- and medium–low-risk areas. In 2020, with the African swine fever epidemic under better control, the risk of pig epidemics in China slowed. Gansu and Shanghai had the greatest epidemic risks and were high-risk and medium–high-risk areas; the risks in other areas were relatively low, and there were 26 low-risk areas.

### 4.2. Discussion

In general, the spatial and temporal variation in pig epidemic risk was the result of multiple factors. This study focused on two aspects: the occurrence and spread of pig epidemics and the spatial and temporal variation in epidemic prevention and control. From the perspective of the occurrence and spread of pig epidemics, the higher the mortality rate was, the faster the spread, the more serious the situation, the greater the risk of pig epidemics, and the more obvious the risk spillover effect to other regions.

Taking African swine fever as an example, the mortality and culling rates caused by the national pig epidemic in 2018–2019 were 41.97% and 0.49% and 40.95% and 0.19%, respectively, much higher than the respective 21.50% and 0.001% in 2017 before the African swine fever outbreak. Among them, the highest culling rate was 7.64% in Beijing in 2018 and 2.16% in Tibet in 2019, which was consistent with the regional distribution and annual changes in the risk of pig epidemics. In terms of spatial and temporal differences in pig epidemic prevention and control, factors such as breeding density, industrial structure, and prevention and control foundation were closely related to the risk of an epidemic; the greater the breeding density was, the more vulnerable the industrial structure, and the poorer the prevention and control foundation was, and the more serious the risk faced by pig breeding. In terms of breeding density, the highest was 10.73 heads/ha in Hainan, and the lowest was 0.72 heads/ha in Inner Mongolia. The risk of pig epidemics in Hainan was obviously higher than that in Inner Mongolia. In terms of industrial structure, the highest was 19.65% in Hunan, the lowest was 1.40% in Tibet, and the risk of pig epidemics in Tibet was higher than that in Hunan. In terms of prevention and control foundation, each township animal husbandry and veterinary station in Hainan Province needed to serve 21,086.40 pigs, while the corresponding indicator in Tibet was only 136.94 pigs. The pressure of prevention and control in Hainan was significantly greater than that in Tibet, so the corresponding risk of pig epidemics in Hainan was higher than that in Tibet.

## 5. Conclusions

On the basis of understanding the situation of pig epidemics, the following research conclusions were obtained by constructing a pig epidemic risk assessment index system and combining it with the entropy method to explore the characteristics of spatial and temporal differentiation of pig epidemic risk in China.

First, the overall trend in pig epidemics first increased and then decreased. African swine fever was the most serious epidemic in recent years, and swine erysipelas, swine pasteurellosis, porcine reproductive and respiratory syndrome, classical swine fever and other pig diseases were also more common. Pig epidemics showed a trend of accelerated spread nationwide after continued weakening and now have only sporadic occurrences in some areas. Although African swine fever and other pig diseases are better controlled, the risks and challenges, which include many types and a wide range of diseases, cannot be ignored. Livestock breeding and epidemic prevention and control capacity were not strong during the study period, and the risk of introduction of foreign animal diseases was greater.

Second, the spatial and temporal variation in pig epidemic risk was obvious. In 2017–2020, there was 1 high-risk area, 2 medium–high-risk areas, 10 medium-risk areas, 17 medium–low-risk areas and 1 low-risk area. The top five provinces in terms of pig epidemic risk were Beijing, Hainan, Liaoning, Tibet and Zhejiang, and the bottom five provinces were Shandong, Hebei, Liaoning, Ningxia and Inner Mongolia. There were significant differences in the regional distribution of epidemic risk in different years, with the high-risk areas being Qinghai, Jiangxi and Chongqing in 2017, Beijing in 2018, Tibet and Jiangsu in 2019, and Gansu in 2020. The temporal and spatial variations in pig epidemics and risks were mainly the result of many reasons, such as the scale of pig breeding, transportation and circulation, epidemic prevention and control capacity in different provinces, and so on.

Based on the above research findings, the following countermeasures are proposed for reference. First, sound monitoring and early warning systems for pig epidemic risk are needed. For domestic and foreign pig and other animal diseases, it is necessary to establish a risk monitoring and early warning system that covers the whole industry chain, including breeding, slaughtering, processing, circulation and sales, so that timely detection, timely reporting, and timely formation and release of epidemic prevention and control early warning plans can provide a reference for scientific decision making for the pig industry and market participants.

Second, the pig epidemic prevention and control system should be improved to include innovations in vaccination and culling, biosecurity and other means of disease prevention and control and should strive for an effective combination of multiple programs and efficient prevention and control. Epidemic detection in the pig industry chain of breeding, transportation, slaughtering and processing should be strengthened. The feeding management system, such as closed feeding and all-in-all-out, should be implemented; the cleaning and disinfection facilities and equipment for epidemic prevention in pig farms should be improved; farm records should be established; and pig identification, such as ear tags with activity, temperature and sound sensors, should be strictly enforced. The strategic planning of animal disease prevention and control science and technology should be strengthened; the promotion of scientific and technological innovation, transformation of results and integrated demonstrations should be accelerated; and research and development of vaccines for African swine fever and other major diseases should be accelerated. The construction of a grassroots animal disease prevention and control system, an epidemic prevention team, and grassroots epidemic prevention capacity should also be strengthened and improved. The culling subsidy system should be improved, and the enthusiasm and initiative of farm households for prevention and control should be fully mobilized.

Third, the regional collaboration mechanism for epidemic prevention and control should be strengthened. To jointly build a solid “defense system” for the high-quality development of the pig industry, it is necessary to establish an overall responsibility-sharing mechanism for the prevention and control of swine diseases in production and marketing areas based on the “national chessboard”; to streamline and consolidate the responsibilities and obligations of various areas; to implement zonal prevention and control and regionally differentiated prevention and control strategies; to compensate for shortcomings and weaknesses according to local conditions; and to strengthen regional cooperation in the prevention and control of the entire industrial chain of pig and product production, transportation and slaughtering.

## Figures and Tables

**Figure 1 vetsci-10-00485-f001:**
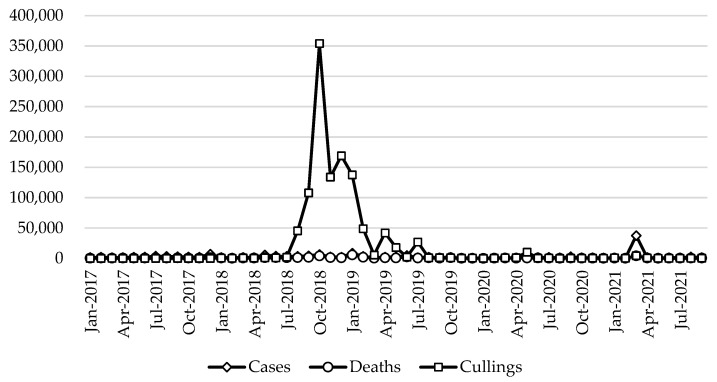
Monthly trends in the pig epidemic (heads), 2017–2021. Data source: Official Veterinary Bulletin, the Ministry of Agriculture and Rural Affairs of the People’s Republic of China.

**Figure 2 vetsci-10-00485-f002:**
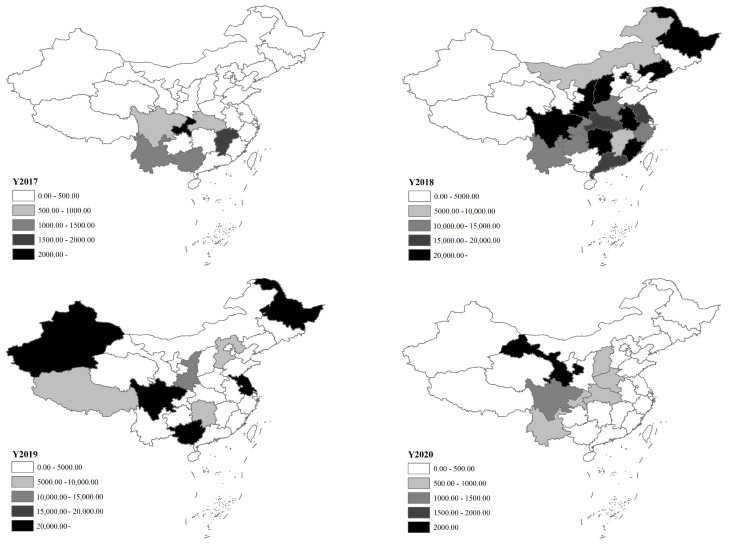
Regional distribution of pig epidemics, 2017–2020. Data source: Official Veterinary Bulletin, the Ministry of Agriculture and Rural Affairs of the People’s Republic of China.

**Figure 3 vetsci-10-00485-f003:**
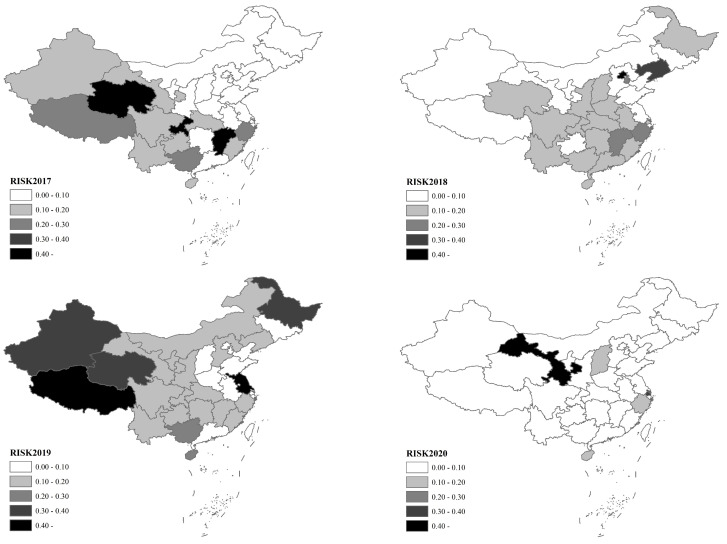
Distribution of the risk of pig epidemics, 2017–2020.

**Table 1 vetsci-10-00485-t001:** Pig epidemic situation, 2017–2021 (heads).

Type		2017	2018	2019	2020	2021
Classical swine fever	Cases	925	2277	101	161	43
	Deaths	312	1299	50	58	22
	Cullings	42	3669	179	269	18
Porcine reproductive and respiratory syndrome	Cases	625	1033	3576	351	2267
	Deaths	323	526	2099	128	644
	Cullings	0	822	1567	121	56
Swine erysipelas	Cases	11,299	10,087	3162	2125	2708
	Deaths	2812	3176	884	393	495
	Cullings	30	2040	445	218	221
Swine pasteurellosis	Cases	18,897	14,948	10,572	9491	41,123
	Deaths	3923	3335	3991	2821	5057
	Cullings	51	2431	1726	1467	4225
African swine fever	Cases	0	8127	12,192	1249	1124
	Deaths	0	5706	8104	978	1008
	Cullings	0	804,248	280,888	12,156	2443
Foot and mouth disease	Cases	67	388	0	40	4
	Deaths	0	2	0	1	4
	Cullings	144	2302	0	248	29

Data source: Official Veterinary Bulletin, the Ministry of Agriculture and Rural Affairs of the People’s Republic of China; the data for 2021 were from January to September.

**Table 2 vetsci-10-00485-t002:** Risk assessment index system for pig epidemics.

Type	Indicator	Unit	Calculation Method
Hazard	Morbidity rate	%	Ratio of cases to pig inventory due to the epidemic
	Mortality rate	%	Ratio of deaths to cases due to the epidemic
	Culling rate	%	Ratio of cullings to pig inventory due to the epidemic
Vulnerability	Breeding density	Heads/ha	Ratio of pig inventory to grain cultivation area
	Industrial structure	%	Ratio of pig industry output to total agricultural output
	Prevention and control foundation	Heads/person	Ratio of pig inventory to the number of staff in the township animal husbandry and veterinary station

**Table 3 vetsci-10-00485-t003:** Results of the risk assessment of pig epidemics.

Province	Risk	Hazard (%)				Vulnerability (Heads/ha, %, Heads/Person)				Risks Properties
		Hazard	Morbidity Rate	Mortality Rate	Culling Rate	Vulnerability	Breeding Density	Industrial Structure	Prevention and Control Foundation	
Beijing	0.682	0.799	0.011	57.589	1.711	0.435	9.312	7.146	665.138	High
Tianjin	0.245	0.216	0.008	48.812	0.271	0.306	4.770	9.643	3564.858	Medium
Hebei	0.104	0.052	0.001	44.684	0.008	0.214	2.666	11.241	3619.070	Medium–low
Shanxi	0.163	0.169	0.004	62.037	0.205	0.148	1.682	9.254	1850.031	Medium–low
Inner Mongolia	0.125	0.127	0.002	86.695	0.040	0.122	0.722	4.973	1091.762	Medium–low
Liaoning	0.337	0.361	0.006	79.015	0.607	0.286	3.515	7.948	4311.222	Medium–high
Jilin	0.112	0.109	0.001	94.091	0.008	0.118	1.546	12.988	1880.681	Medium–low
Heilongjiang	0.210	0.239	0.011	75.542	0.191	0.147	0.933	8.784	2933.775	Medium
Shanghai	0.260	0.178	0.013	32.667	0.119	0.435	6.905	8.685	5227.565	Medium
Jiangsu	0.181	0.159	0.007	42.542	0.163	0.227	2.361	6.062	3178.426	Medium–low
Zhejiang	0.268	0.188	0.016	40.561	0.053	0.438	5.389	6.182	7218.300	Medium
Anhui	0.143	0.095	0.003	47.968	0.065	0.244	1.809	13.809	7132.609	Medium–low
Fujian	0.263	0.120	0.003	31.824	0.192	0.566	9.852	6.784	5339.400	Medium
Jiangxi	0.268	0.266	0.031	24.120	0.011	0.271	3.871	11.822	4223.106	Medium
Shandong	0.097	0.006	0.000	11.930	0.004	0.289	3.329	9.222	5167.191	Low
Henan	0.167	0.052	0.001	42.100	0.009	0.412	3.646	11.618	10,664.556	Medium–low
Hubei	0.172	0.109	0.011	19.854	0.023	0.305	4.685	13.005	4659.298	Medium–low
Hunan	0.165	0.074	0.004	22.738	0.060	0.358	7.447	19.646	4234.683	Medium–low
Guangdong	0.159	0.017	0.001	7.127	0.027	0.460	8.356	9.164	3984.721	Medium–low
Guangxi	0.257	0.170	0.015	31.939	0.061	0.439	7.550	9.288	4472.878	Medium
Hainan	0.367	0.086	0.004	48.855	0.010	0.962	10.731	6.892	21,086.396	Medium–high
Chongqing	0.234	0.218	0.021	44.552	0.030	0.269	5.420	13.460	1967.587	Medium
Sichuan	0.176	0.118	0.009	34.724	0.041	0.300	6.116	14.810	2537.777	Medium–low
Guizhou	0.141	0.071	0.002	47.281	0.022	0.289	5.047	11.799	2991.554	Medium–low
Yunnan	0.190	0.105	0.008	38.250	0.009	0.370	6.924	16.610	4722.945	Medium–low
Tibet	0.282	0.329	0.020	20.625	0.415	0.183	2.202	1.402	136.941	Medium
Shaanxi	0.165	0.140	0.005	43.802	0.162	0.219	2.777	8.630	2868.769	Medium–low
Gansu	0.160	0.153	0.010	53.074	0.055	0.174	2.092	5.427	1123.462	Medium–low
Qinghai	0.248	0.281	0.021	89.753	0.042	0.178	2.360	4.014	434.220	Medium
Ningxia	0.118	0.104	0.003	61.702	0.047	0.149	1.130	3.150	1031.313	Medium–low
Xinjiang	0.170	0.176	0.005	42.524	0.246	0.157	1.521	2.798	571.617	Medium–low

Note: This study classified risk into five levels: low, medium–low, medium, medium–high, and high risk, based on risk indices of <0.10, 0.10–0.20, 0.20–0.30, 0.30–0.40, and >0.40, respectively.

## Data Availability

Data are available on request from the authors.

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
