# Peer review of "Risk Assessment and Response Strategy for Pig Epidemics in China"

_vetsci, 2023, doi:10.3390/vetsci10080485_

Round 1
Reviewer 1 Report
This is a very lengthy paper on a very important topic that must be shortened to in order to engage the reader. My main concern with this paper is the data presented in part 2, which is orgifginated from the Goverment of China. I think the data are skewed and biased. I am willing to reconsider this paper after complytley removing part 2.
No special comments
Author Response
Dear editor and reviewers,
We appreciate your letters and valuable comments and suggestions. We have carefully revised and improved the paper based on your comments and suggestions. However, we are very sorry to say that we do not agree with some of the reviewers' comments and suggestions. We have provided appropriate explanations and clarifications and hope that you will forgive us. We believe that it is possible for academic research to have different opinions and ideas. And our study is only a throwaway for those authors who are interested in China's pig industry and epidemics. We respond to your comments and suggestions as follows:
This is a very lengthy paper on a very important topic that must be shortened to in order to engage the reader. My main concern with this paper is the data presented in part 2, which is orgifginated from the Goverment of China. I think the data are skewed and biased. I am willing to reconsider this paper after complytley removing part 2.
Dear reviewer, thank you for your valuable comments and suggestions on this paper. The data for this study are from the Official Veterinary Bulletin published by the Ministry of Agriculture and Rural Affairs of China, which is an authoritative data published by the authority. We can also understand your thoughts because many people are skeptical about the data published by the Chinese government. We do not deny that the data may be skewed and biased, but for the time being, this data is more reliable because we have no way to find a second source of data. At the same time, this data shows the trend and regional distribution of animal epidemics, which is generally consistent with the actuality of outbreaks in China. Therefore, we retained Part 2. However, since other reviewers also suggested revising and improving the content of Part 2. We have made a lot of deletions and streamlined the contents of Part 2, and hope to earn your forgiveness. Of course, if Part 2 does need to be deleted, we agree to do so.
Reviewer 2 Report
The manuscript presents official data on pigs culled in different provinces in China, and this data is utilized to calculate different indices for risk assessment of an epidemic. Overall the manuscript could be improved to simplify its goals and emphasize its main results. Below are some suggestions to consider:
1) The introduction is long and needs to be organized better to convey the main points. There are several places where the text deviates from the main point. For example, ASF and biosecurity are talked about much in Introduction, but biosecurity is not covered much in later sections.
2) Section 2 could be removed or shortened because the statements and section headings are strong and may not reflect everything correctly, such as Fig 1.
3) The inferences from the epidemics from 2017-2021 could be described in more depth besides listing the official data. For example, why the epidemic trends are as shown in figure (increased then decreased) and any suggestions?
4) The countermeasures used during the epidemics could be described in more detail. This is briefly touched on page 13 using early warning systems, sound monitoring systems etc. Could you also list some examples of countermeasures or biosecurity tools deployed in swine farms, transport, and slaughterhouses to detect epidemics? One such technology is using activity, temperature and sound sensors in ear tags of pigs.
5) The conclusions can be revised. The reasoning behind the spatial and temporal variations in the pig epidemics within different provinces could be described, beyond stating the statistics. Some of the text could be moved to the Discussion section from Conclusion and Section 2.
6) How do you justify or validate your indexing system? In other words, how do you know these indices are sufficient and accurately show the risks of epidemic?
Overall, your aspects on health monitoring of pigs needs to be made stronger, because it is only mentioned in the conclusion section in a few lines.
the quality of English is fine.
Author Response
Dear editor and reviewers,
We appreciate your letters and valuable comments and suggestions. We have carefully revised and improved the paper based on your comments and suggestions. However, we are very sorry to say that we do not agree with some of the reviewers' comments and suggestions. We have provided appropriate explanations and clarifications and hope that you will forgive us. We believe that it is possible for academic research to have different opinions and ideas. And our study is only a throwaway for those authors who are interested in China's pig industry and epidemics. We respond to your comments and suggestions as follows:
1. The introduction is long and needs to be organized better to convey the main points. There are several places where the text deviates from the main point. For example, ASF and biosecurity are talked about much in Introduction, but biosecurity is not covered much in later sections.
Dear reviewer, thank you for your valuable comments and suggestions. We have revised and improved the introduction by removing most of the description of biosecurity and combining the first and second paragraphs into one.
2. Section 2 could be removed or shortened because the statements and section headings are strong and may not reflect everything correctly, such as Fig 1.
Dear reviewer, thank you for your valuable comments and suggestions. We have drastically compressed and streamlined the content of Section 2 and refined the relevant headings. In addition, Figure 1 is plotted based on monthly data, which we believe provides a better representation of the temporal characteristics of pig epidemics, and we have retained it.
3. The inferences from the epidemics from 2017-2021 could be described in more depth besides listing the official data. For example, why the epidemic trends are as shown in figure (increased then decreased) and any suggestions?
Dear reviewer, thank you for your valuable comments and suggestions. Based on your comments and suggestions, after describing the temporal variations of pig epidemics from 2017-2021, we briefly explain why pig epidemics are increasing and then decreasing. As for the recommendations, we have uniformly placed them all at Conclusion. And we hope to earn your forgiveness.
4. The countermeasures used during the epidemics could be described in more detail. This is briefly touched on page 13 using early warning systems, sound monitoring systems etc. Could you also list some examples of countermeasures or biosecurity tools deployed in swine farms, transport, and slaughterhouses to detect epidemics? One such technology is using activity, temperature and sound sensors in ear tags of pigs.
Dear reviewer, thank you for your valuable comments and suggestions. Based on your comments and suggestions, we have added to the second paragraph of the policy proposal actions that should be taken during a pig epidemic. The main components are: Epidemic detection in the pig industry chain of breeding, transportation, slaughtering and processing should be strengthened. The feeding management system such as closed feeding and all-in-all out should be implemented; the cleaning and disinfection facilities and equipment for epidemic prevention in pig farms should be improved; the farm records should be established; and the pig identification, such as ear tags with activity, temperature and sound sensors, should be strictly enforced.
5. The conclusions can be revised. The reasoning behind the spatial and temporal variations in the pig epidemics within different provinces could be described, beyond stating the statistics. Some of the text could be moved to the Discussion section from Conclusion and Section 2.
Dear reviewer, thank you for your valuable comments and suggestions. Based on your comments and suggestions, we have discussed the reasons behind the spatial and temporal variations of pig epidemics in different provinces in Conclusion. It should be noted that the discussion section is mainly based on the results of our evaluation, so in this paper we base our discussion only on the selected indicators, which is more relevant. And we hope you will forgive us.
6. How do you justify or validate your indexing system? In other words, how do you know these indices are sufficient and accurately show the risks of epidemic?
Dear reviewer, thank you for your valuable comments and suggestions. In this paper, we assess the risk of pig epidemic by constructing an indicator system and combining the entropy method. Our principles for selecting indicators are scientific and accessible, and we explain the selection of indicators and the relationship between these indicators and the risk in Methods. The results of pig epidemic risk assessment, that is, the spatial and temporal characteristics of the risk, are generally consistent with the situation of pig epidemics in China and can be corroborated with relevant data and information. We believe that our study is sound and of reference value. Of course, if other readers are interested, they can further expand on the basis of our study.
7. Overall, your aspects on health monitoring of pigs needs to be made stronger, because it is only mentioned in the conclusion section in a few lines.
Dear reviewer, thank you for your valuable comments and suggestions. Based on your comments and suggestions, we have added pig health monitoring to Conclusion. As mentioned above, it is in the second paragraph of the policy recommendations. The Part 2 and Part 3 focus on analyzing the characteristics and risk of pig epidemics and assessing the risk, and we believe that the target content should be better presented in these two parts, which may be less related to the issue of pig health monitoring.
Reviewer 3 Report
In this work entitled "Risk Assessment and Response Strategy for Pig Epidemics in China," the authors analyze nationally available data on swine morbidity, mortality, culling, and industry infrastructure between the years of 2017 and 2020 to assess the changes in epidemic risk over the period of a major disease outbreak. The work offers interesting methodology to assess risk and hazard scores, which may guide implementation of future countermeasures.
Major comments:
1. The Introduction (section 1, lines 63-110) could use major revision with respect to discussion of the existing literature. Many works are cited but have minimal summary or discussion that link their conclusions to the work performed in the manuscript. It may benefit the work to provide a more direct link between the literature introduced here and the analyses performed by the authors.
2. The values in Table 3 appear to be calculated across the entire study period (2017-2020), however the discussion focuses on temporal differences on a yearly basis. A clear separation or indication of which data are used to calculate which values would increase the readability of this section (i.e. data used to determine risk classification versus data used to perform the spatial and temporal analyses).
3. In the discussion of spatial and temporal results (lines 408-414), the authors state a relationship between the study indicators and risk/hazard. The authors have calculated aggregate risk/hazard scores, however the individual contribution of any one indicator and the relative importance of each indicator with respect to the other indicators are not directly assessed. From a disease prevention and control standpoint, understanding which of these indicators to prioritize seems like useful information and would more strongly support the countermeasures recommendations proposed in the conclusions. Is there a correlation or importance metric that could be assessed for the indicators?
Minor comments:
1. Lines 111-112: "In general, a series of studies on animal disease risk and assessment have been conducted, resulting in many findings with important reference." The intention of this statement is not clear.
2. Lines 232-233: "Data from the World Organisation for Animal Health (OIE)..." The abbreviation for this organization has officially changed from OIE to WOAH.
Grammatical errors were minor.
Author Response
Dear editor and reviewers,
We appreciate your letters and valuable comments and suggestions. We have carefully revised and improved the paper based on your comments and suggestions. However, we are very sorry to say that we do not agree with some of the reviewers' comments and suggestions. We have provided appropriate explanations and clarifications and hope that you will forgive us. We believe that it is possible for academic research to have different opinions and ideas. And our study is only a throwaway for those authors who are interested in China's pig industry and epidemics. We respond to your comments and suggestions as follows:
1. The Introduction (section 1, lines 63-110) could use major revision with respect to discussion of the existing literature. Many works are cited but have minimal summary or discussion that link their conclusions to the work performed in the manuscript. It may benefit the work to provide a more direct link between the literature introduced here and the analyses performed by the authors.
Dear reviewer, thank you for your valuable comments and suggestions. It is difficult to find literature relevant to this research because there is very little literature discussing our research topic. At present, the literature we could find focuses more on pig epidemic risk, transmission characteristics, and animal disease prevention and control, etc., and these literatures do not bother to discuss the spatial and temporal characteristics of risk. Therefore, the logic of our layout in the literature review is to first review the assessment methods of pig epidemic risk, then discuss the risk and impact of pig epidemics transmission, and finally summarize the relevant measures to cope with the risk of pig epidemics. In our opinion, all these contents are related to the later work, because the contents of the later work are also mainly focused on the aspects of risk assessment and response measures. We would appreciate your understanding of the shortcomings in the literature review. In addition, we have adapted and refined the literature review as appropriate.
2. The values in Table 3 appear to be calculated across the entire study period (2017-2020), however the discussion focuses on temporal differences on a yearly basis. A clear separation or indication of which data are used to calculate which values would increase the readability of this section (i.e. data used to determine risk classification versus data used to perform the spatial and temporal analyses).
Dear reviewer, thank you for your valuable comments and suggestions. The results in Table 3 are calculated over the entire study period (2017-2020) and are based on four-year averages. The results in Figure 3 are calculated based on data for each year. In addition, we have analyzed the results in both Table 3 and Figure 3 in depth and have not focused our discussion on temporal differences on the yearly basis. Both are the focus of this study. A clear separation or indication of which data are used to calculate which values was explained in the first paragraph of Results.
3. In the discussion of spatial and temporal results (lines 408-414), the authors state a relationship between the study indicators and risk/hazard. The authors have calculated aggregate risk/hazard scores, however the individual contribution of any one indicator and the relative importance of each indicator with respect to the other indicators are not directly assessed. From a disease prevention and control standpoint,
understanding which of these indicators to prioritize seems like useful information and would more strongly support the countermeasures recommendations proposed in the conclusions. Is there a correlation or importance metric that could be assessed for the indicators?
Dear reviewer, thank you for your valuable comments and suggestions. The logic of our study is to first select individual indicators and then construct a system of evaluation indicators to assess the risk of pig epidemics, and our result is to obtain the aggregate risk/hazard scores. The individual contribution of each indicator and the relative importance of each indicator is not the focus of this paper. Of course, when calculating the weights using the entropy method, it is possible to arrive at the issue you mention. That is: the hazard and vulnerability weights calculated by the entropy method for 2017-2020 were 0.679 and 0.321, respectively; the hazard indicators were 0.243, 0.105 and 0.652 for morbidity, mortality and culling rates, respectively; and the vulnerability indica-tors were 0.393, 0.125 and 0.482 for breeding density, industrial structure and prevention and control foundation, respectively. The weights of each indicator reflected that the risk of a pig epidemic came more from hazards; the culling rate regarding hazard factors was more important; and the foundation of prevention and control of vulnerability factors was more important. As for specifically addressing the degree of contribution and importance of individual indicators, we will discuss this systematically in our future research. We hope you will forgive us.
4. Lines 111-112: "In general, a series of studies on animal disease risk and assessment have been conducted, resulting in many findings with important reference." The intention of this statement is not clear.
Dear reviewer, thank you for your valuable comments and suggestions. Based on your comments and suggestions, we have removed this sentence "In general, a series of studies on animal disease risk and assessment have been conducted, resulting in many findings with important reference".
5. Lines 232-233: "Data from the World Organisation for Animal Health (OIE)..." The abbreviation for this organization has officially changed from OIE to WOAH.
Dear reviewer, thank you for your valuable comments and suggestions. Based on your comments and suggestions, we have changed the OIE to WOAH.
Reviewer 4 Report
In my opinion it's complicate to judge this paper without understanding of the pig production system and veterinary service in China. In general, the epidemiology of disease, properties of infection agent (for example, highly pathogenic PRRS and "regular" PRRS) play a key role in risk assessment. Number of small backyard farms with the smaller number of animals pose higher risk than one industrial farm with bigger number of pigs. Also authors do not take in to account very specific Chinese pork market with requirement of chilled but not frozen pork and very long transportation routes for animal slaughtering. These are just some of the questions to this paper.
From another hand, very little information about the situation with pig diseases in China is available, and it is useful to know the opinion of national epidemiologists on risk assessment.
I do not think that this paper can be improved and see no reason to reject it. I rely on the editor's decision. My personal opinion that the manuscript does not bring in to much science, but can be interesting for readers as it's describe the situation with pig diseases in China.
English is fine for me, several typos present in the text, but nothing serious.
Author Response
Dear editor and reviewers,
We appreciate your letters and valuable comments and suggestions. We have carefully revised and improved the paper based on your comments and suggestions. However, we are very sorry to say that we do not agree with some of the reviewers' comments and suggestions. We have provided appropriate explanations and clarifications and hope that you will forgive us. We believe that it is possible for academic research to have different opinions and ideas. And our study is only a throwaway for those authors who are interested in China's pig industry and epidemics. We respond to your comments and suggestions as follows:
In my opinion it's complicate to judge this paper without understanding of the pig production system and veterinary service in China. In general, the epidemiology of disease, properties of infection agent (for example, highly pathogenic PRRS and "regular" PRRS) play a key role in risk assessment. Number of small backyard farms with the smaller number of animals pose higher risk than one industrial farm with bigger number of pigs. Also authors do not take in to account very specific Chinese pork market with requirement of chilled but not frozen pork and very long transportation routes for animal slaughtering. These are just some of the questions to this paper.
From another hand, very little information about the situation with pig diseases in China is available, and it is useful to know the opinion of national epidemiologists on risk assessment.
I do not think that this paper can be improved and see no reason to reject it. I rely on the editor's decision. My personal opinion that the manuscript does not bring in to much science, but can be interesting for readers as it's describe the situation with pig diseases in China.
Dear reviewer, thank you for your valuable comments and suggestions. This article is based on the perspective of social science to discuss the issue of pig epidemics, on the epidemiology of disease, properties of infection agent and other characteristics, more from the natural science to discuss. At present, the only data we can get from the social science perspective are morbidity rate, mortality rate, etc., and other data are hard to come by. Therefore, we conducted the present study based on the present design. Also, there is relevant content in the paper "Number of small backyard farms with the smaller number of animals pose higher risk than one industrial farm with bigger number of pigs". We have also added the issue of the peculiarities of China's pork consumption market, including the population's preference for hot fresh meat, long-distance transportation and slaughter, when we discuss the risks and challenges of pig epidemics in Part 2. Of course, there are some shortcomings in our study, but the main purpose is to provide a reference for those readers who are interested in China's pig industry and epidemics. Thank you for understanding our work.
In addition, we have thoroughly revised the full text and improved the English language.
Thanks again for your comments and suggestions. Please forgive us for not revising the article based on some of the comments and suggestions, and we will improve it in future studies.
Round 2
Reviewer 2 Report
Thank you for your edits.
Reviewer 3 Report
The authors have addressed my primary concerns in the manuscript revision.